# Two subunits of human ORC are dispensable for DNA replication and proliferation

Etsuko Shibata, Manjari Kiran, Yoshiyuki Shibata, Samarendra Singh, Shashi Kiran, Anindya Dutta*

Department of Biochemistry and Molecular Genetics, University of Virginia School of Medicine, Charlottesville, United States

**Abstract** The six-subunit Origin Recognition Complex (ORC) is believed to be an essential eukaryotic ATPase that binds to origins of replication as a ring-shaped heterohexamer to load MCM2-7 and initiate DNA replication. We have discovered that human cell lines in culture proliferate with intact chromosomal origins of replication after disruption of both alleles of *ORC2* or of the ATPase subunit, *ORC1*. The *ORC1* or *ORC2*-depleted cells replicate with decreased chromatin loading of MCM2-7 and become critically dependent on another ATPase, CDC6, for survival and DNA replication. Thus, either the ORC ring lacking a subunit, even its ATPase subunit, can load enough MCM2-7 in partnership with CDC6 to initiate DNA replication, or cells have an ORC-independent, CDC6-dependent mechanism to load MCM2-7 on origins of replication

## Introduction

The discovery of the six-subunit ORC (*Bell and Stillman, 1992*) identified the long sought initiator protein that binds to replicator sequences to initiate DNA replication in eukaryotes. ORC is an essential six-subunit, ring-shaped ATPase complex that recruits and co-operates with the CDC6 protein to promote the loading of CDT1 and then the MCM2-7 subunits of the replicative helicase during the 'licensing' of origins of replication (*Bleichert et al., 2015*; *Yeeles et al., 2015*; *Bell and Stillman, 1992*; *Blow and Tada, 2000*; *Masai et al., 2010*). Of the six subunits of human ORC, ORC2-5 form a tight core complex (*Dhar et al., 2001*). ORC1 is the only subunit responsible for the ATPase activity of ORC (*Chesnokov et al., 2001*; *Giordano-Coltart et al., 2005*; *Klemm et al., 1997*) All six ORC subunit genes are essential for the viability of *S. cerevisiae* and *S. pombe* (*Bell et al., 1993*; *Foss et al., 1993*; *Herskowitz, 1993*; *Loo et al., 1995*; *Micklem et al., 1993*). It is therefore expected that eukaryotic cells will not be viable, and will not replicate, if any of the ORC subunit genes are deleted. However, we have now deleted both alleles of human *ORC2* or of *ORC1*, to discover that cells can still survive and replicate in the complete absence of either of these two critical subunits of Origin Recognition Complex.

## Results

### Biallelic disruption of the *ORC2* gene

CRISPR/Cas9 was used to insert a ~600 bp blasticidin gene and poly A site into exon 4 (amino acid 40) of *ORC2* in HCT116 p53-/- (*Bunz et al., 1998*) colon cancer cells (WT: HCT116 *p53-/-*, *ORC2+/+*) (*Figure 1A and B*). All clones were viable and proliferated for months despite the disruption of the *ORC2* gene. Immunoblotting and immunoprecipitation-immunoblotting of cell extracts showed that the *ORC2-/-* B2 and BP8 clones had no detectable ORC2 protein (*Figure 1D–H*). An anti-ORC2C

*For correspondence: ad8q@
eservices.virginia.edu

**Competing interests:** The
authors declare that no
competing interests exist.

**Reviewing editor:** Kevin Struhl,
Harvard Medical School, United
States

**eLife digest** Most of the DNA in human cells is packaged into structures called chromosomes. Before a cell divides, the DNA in each chromosome is carefully copied. This process begins at multiple sites (known as origins) on each chromosome. A group of six proteins collectively known as the Origin Recognition Complex (or ORC for short) binds to an origin and then recruits several additional proteins. When the cell is ready, the assembled proteins are activated and DNA copying begins. It is thought that all of the ORC proteins are essential for cells to survive and copy their DNA.

Here, Shibata et al. reveal that human cells can survive without ORC1 or ORC2, two of the six proteins in the ORC complex. Disrupting the genes that encode the ORC1 and ORC2 proteins in human cancer cell lines had little effect on the ability of the cells to copy their DNA and survive. Furthermore, these cells spend the same amount of time copying their DNA and use a similar set of origins as normal cells.

However, the experiments also reveal that cells without ORC1 or ORC2 are more dependent on the presence of one particular protein recruited to the origin after the ORC assembles. Reducing the availability of this protein, CDC6, decreased the ability of these cells to survive and divide. Future efforts will aim to identify the mechanism by which cells bring together the proteins required to copy DNA in the absence of a complete ORC.

antibody recognizing the C terminal half of ORC2 (*Figure 1C*) ensured that a C-terminal fragment of ORC2 was not being expressed from an alternatively spliced transcript using an internal methionine (*Figure 1E*). Quantitative immunoblotting showed that if any ORC2 was expressed, it must be at a level <1% of wild type levels (*Figure 1D*). Quantitative immunoblotting of cell lysates and carefully measured amounts of recombinant bacterially produced ORC2 fragment showed that wild type cells express ~153,000 ORC2 molecules per cell (*Figure 1I,J*). Thus, if the B2 or BP8 clones contain any ORC2 molecules below the level of detection, they can have no more than 1530 molecules/cell. *ORC2* was also deleted in 293T human embryonic kidney cells or HBEC human bronchial epithelial cells immortalized with CDK4 and hTERT, and these cells too continued proliferating in the absence of ORC2 protein (*Figure 1K*).

## CDC6, CDT1 and MCM2-7 loading on chromatin in the absence of ORC2

The *ORC2-/-* cells suffer a decrease of ORC3, ORC4 and ORC5, which are destabilized when not complexed with ORC2 (*Figure 1G*). ORC1 was also decreased, but ORC6, CDC6 and CDT1 were unchanged. There was no activation of the DNA damage checkpoint, measured by the phosphorylation of Chk1 or H2AX, as would have occurred with impaired DNA replication. Chromatin association of ORC2-5 and of ORC1 was decreased in *ORC2-/-* cells (as expected from the decrease of these proteins in cell lysates) (*Figure 1H*). The chromatin association of MCM3, 5 and 7 of the MCM2-7 was reduced, but not completely eliminated. Surprisingly, chromatin association of ORC6 or CDT1 was relatively unchanged, while CDC6 association was slightly increased.

### Initiation of DNA replication in *ORC2-/-* cells

The proliferation rate of the *ORC2-/-* cells was >50% that of WT cells (*Figure 2A*) and ORC2 did not re-appear even after passage of the cells for over a year. The *ORC2-/-* cells did not accumulate in S phase (*Figure 2B*) and completed DNA replication after release from an early S-phase block in the same time as WT cells (*Figure 2C*). The percentage of cells synthesizing DNA during a 30 min pulse was not significantly decreased (*Figure 2D*). By molecular combing the median distance between bi-directional origins of replication in *ORC2-/-* cells was marginally increased to 113–118 kb from 96 kb and the fork progression rate was slightly increased to 1.3–1.5 kb/min from 1.2 kb/min (*Figure 2E*). Given the total DNA content of six billion bp in these cells, this measurement suggests that the *ORC2-/-* cells fire about 52,000 origins of replication.

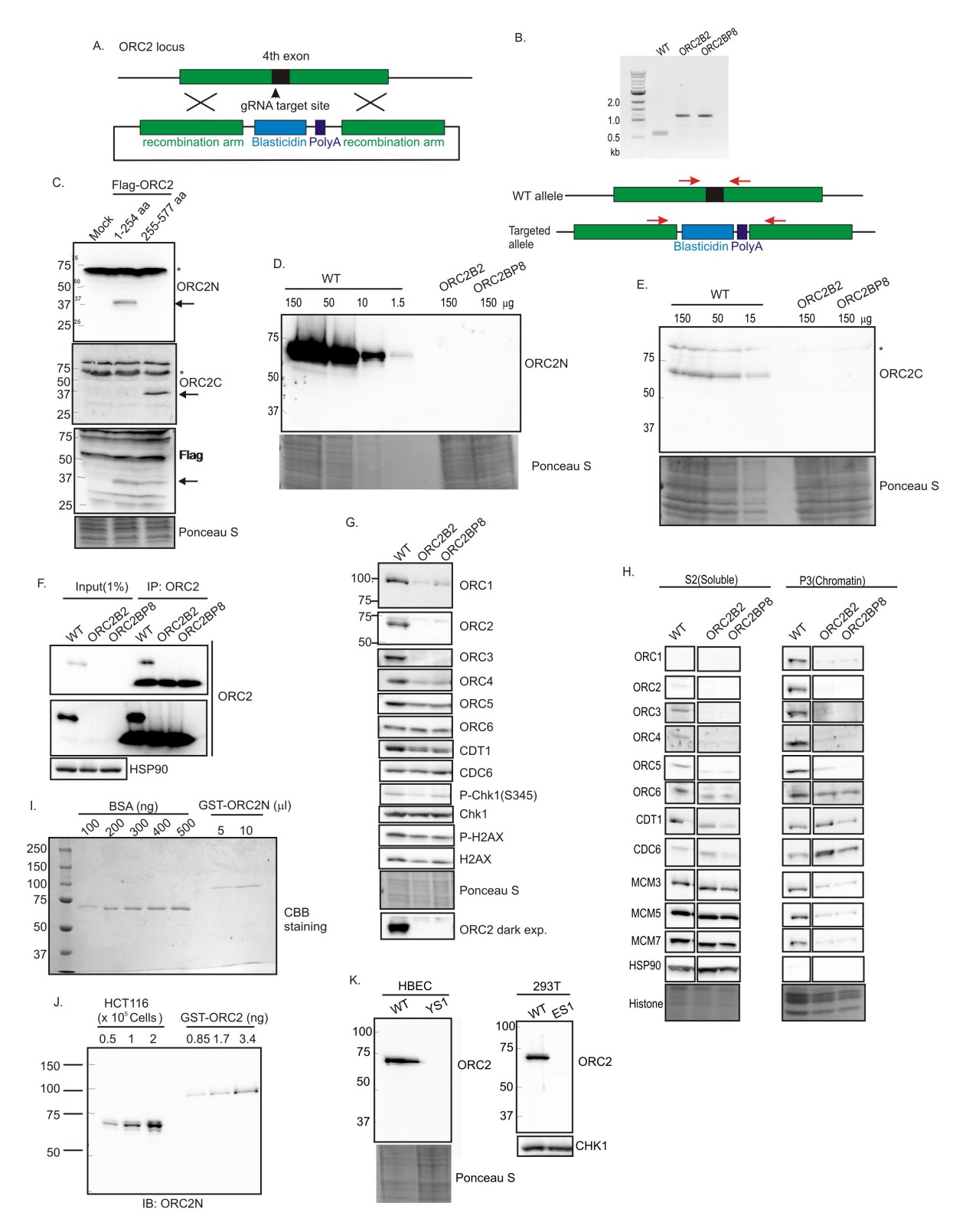

**Figure 1.** Knockout of ORC2 in HCT116 p53-/- cells. (**A**) Strategy for insertion of a blasticidin gene and poly A site in the fourth exon of ORC2 at aa 40 of ORC2. (**B**) PCR on genomic DNA of indicated clones. WT: HCT116 p53-/- and ORC2+/+. ORC2 Knockout clones, B2 and BP8 have an insert on both alleles of ORC2 as indicated by the absence of 0.6 kb PCR product. (**C**) Verification of antibodies recognizing N-terminal or C-terminal parts of ORC2. Recombinant ORC2 protein halves with Flag epitope tags were expressed and blotted with indicated antibodies. Ponceau S staining of total protein

*Figure 1 continued on next page*

*Figure 1 continued*

shows equal loading of lanes.* indicates full length endogenous ORC2 protein. Arrow indicates recombinant protein. (**D**) Quantitative Western blot for ORC2 with an antibody recognizing the N-terminal half of ORC2. Indicated amount of lysate loaded in each lane. (**E**) Western blot with antibody recognizing C-terminal half of ORC2. * Non specific band (**F**) Input cell lysate and immunoprecipitates of ORC2 immunoblotted for ORC2. Darker exposure of the top blots is shown in the middle. HSP90 in the cell lysate or the IgG band in the immunoprecipitate serves as loading control. (**G**) Western blot for indicated proteins in clones indicated on the top. Darker exposure of the ORC2 blots is shown at the bottom. Ponceau S stains all proteins on the blot and also indicates equal loading of lanes. (**H**) Immunoblot of soluble and chromatin-associated proteins in the clones indicated at the top. Ponceau S staining of histones serves as loading control for chromatin fractions. For each panel, all the lanes are from the same blot and exposure. (**I**) Comparison of Coomassie Brilliant Blue signal of pure BSA and recombinant purified GST-ORC2 to show that the top-most band in the ORC2 lane is at 170 ng/ 10 μl. (**J**) Immunoblot with different amounts of cell lysate with the GST-ORC2 to show that 1×10e5 cells give an ORC2 signal equal to 2.54 ng (1.4 fold of 1.67 ng) of GST-ORC2, which corresponds to 153×10e8 molecules of GST-ORC2. (**K**) Western blot of ORC2 in HBEC and 293T cell lines. Ponceau S staining of total protein or immunoblot of Chk1 show equal loading of the pairs of lanes.

We mapped replication initiation sites by a second method: enriching for BrdU-labeled, origin-centered nascent strands by immunoprecipitation and sequencing (BrIPseq) (*Karnani et al., 2010*). ~13,000 BrIPseq origins were mapped to unique DNA sequence in the *ORC2-/-* cells compared to ~20,000 in WT cells (*Figure 2F*). Taking into consideration repetitive DNA and the diploid nature of the human genome, this method also suggests that about 52,000 origins are fired in the *ORC2-/-* cells. 40% of the BrIP-seq peaks in *ORC2-/-* cells overlapped with that in WT cells, comparable to the overlap reported among origins mapped by different groups (*Karnani et al., 2010*). The <100% of overlap is explained by plasticity of origin usage (*Cadoret et al., 2008*). The inter-origin distances for the BrIPseq origins in *ORC2-/-* cells (26 kb) were slightly longer than in the WT (25 kb) cells (P value ~4.3e-07, Wilcoxon rank sum test) and the chromosome-by-chromosome distribution of inter-origin distances was similar to that in WT cells (*Figure 2G*). The 4-fold shorter inter-origin distance in a population-based approach of origin mapping (BrIPseq) compared to molecular combing is also due to the plasticity of origin usage: only one out of four possible origins in a single DNA segment fire in one S phase, while all four origins are used in the entire population of cells (see discussion in *Karnani et al., 2010*). Origins are enriched in gene-rich domains and near transcription start sites (*Figure 2H and I*) as reported in previous studies (*Karnani et al., 2010*; *Cadoret et al., 2008*; *Danis et al., 2004*; *MacAlpine et al., 2004*; *Mesner et al., 2011*; *Sequeira-Mendes et al., 2009*).

## Survival of *ORC1-/-* cells

We next disrupted the only subunit of ORC demonstrated to have ATPase activity, ORC1 (*Chesnokov et al., 2001*; *Giordano-Coltart et al., 2005*; *Klemm et al., 1997*). A blasticidin or a plasmid-derived DNA fragment was inserted in exon 1 of *ORC1* at the initiator methionine (*Figure 3A,B*). Western blotting and immunoprecipitation-western blotting of ORC1 protein showed no ORC1 protein in the *ORC1-/-* clones (*Figure 3C–E*). ORC1 is loosely associated with the other subunits of ORC, and unlike ORC2, ORC1 depletion did not decrease ORC2, ORC3, ORC4, ORC5, ORC6, CDT1, and CDC6 (*Figure 3C*). Although ORC1 was absent, the remaining five subunits of ORC along with CDC6 and CDT1 (at least in one clone) could associate with chromatin. However, the chromatin loading of MCM3, 5, and 7 subunits of MCM2-7 is significantly decreased (*Figure 3D*).

## DNA replication in *ORC1-/-* clones

At early passage (at one month), the *ORC1-/-* clones proliferated at 10–20% of the rate of WT cells but by 6 months of passage, their proliferation rates at about 50% of WT cells without reappearance of ORC1 protein (*Figures 4A* and *3E*). The cells did not accumulate in S phase (*Figure 4B*) and passage through S phase was not slowed (*Figure 4C*). By molecular combing, inter origin distance in *ORC1-/-* cells was unchanged or slightly decreased (*Figure 4D*). Fork progression was decreased to 0.8–1.0 kb/min from 1.2 kb /min. BrIPseq analysis showed that the *ORC1-/-* cells fire ~13,000 unique origins, with 43% of them overlapping with origins in WT cells (*Figure 4E*). The distribution of inter-origin distances between chromosomes, enrichment of origins in gene-rich segments and near transcription start sites was similar to that of WT cells or *ORC2-/-* cells (*Figure 4F–H*).

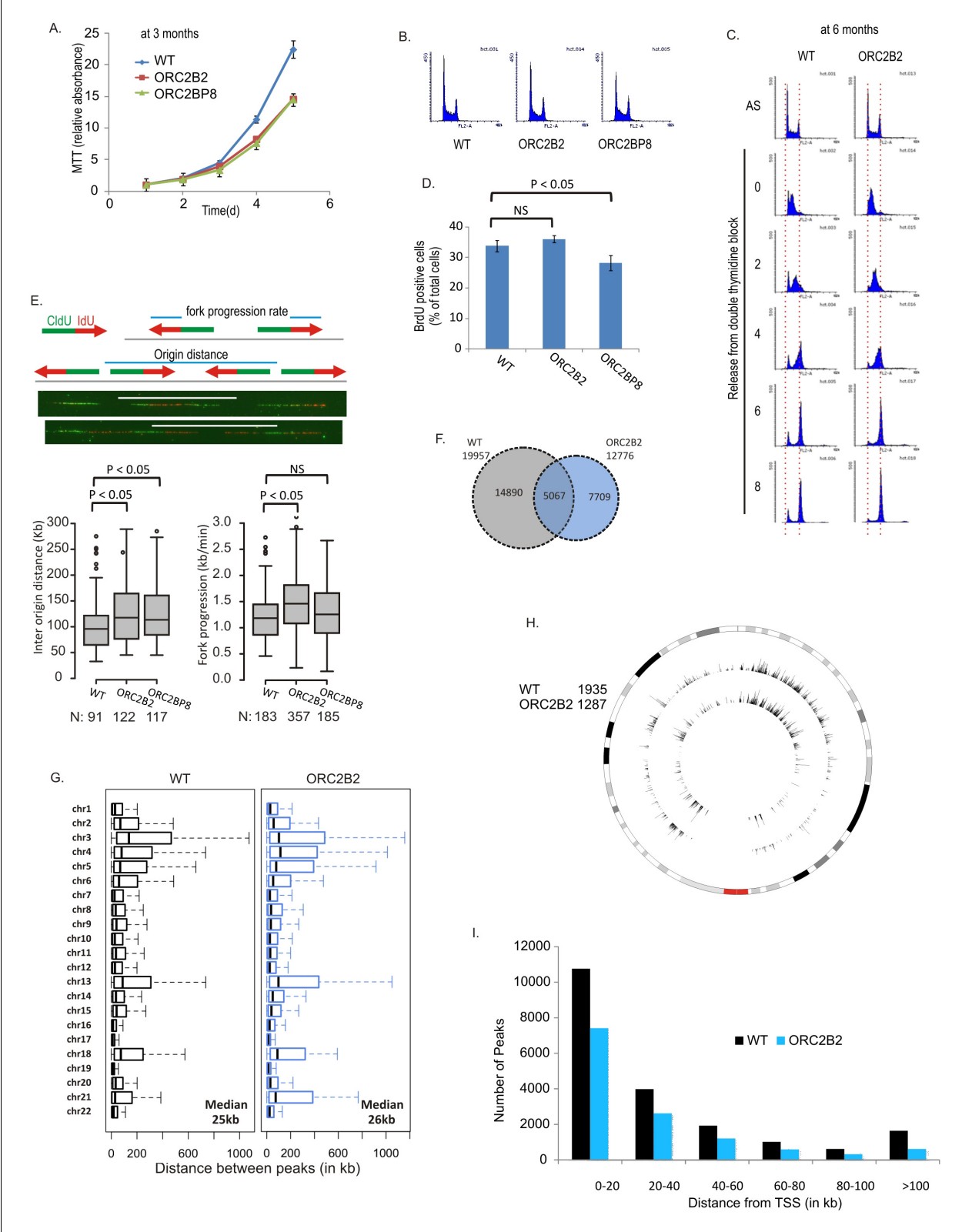

**Figure 2.** Cell proliferation and DNA replication in the ORC2-/- cell lines. (**A**) Growth curves of indicated clones of cells over five days, expressed as MTT absorbance relative to the level at day 1. (Mean ± S.D.; n = 4 biological replicates. Cells after passage for three months). (**B**) FACS profile of propidium-iodide stained cell-cycle asynchronous cells from indicated clones. (**C**) Cells arrested in double-thymidine block released into nocodazole containing medium and harvested at indicated times after release to measure rate of progression through S phase. AS: asynchronous cells. The red

*Figure 2 continued on next page*

*Figure 2 continued*
dotted lines indicate cells with G1 and G2 DNA content. (D) Cell-cycle asynchronous cells labeled with BrdU for 30 min. % of BrdU labeled cells evaluated by two color FACS. (Two-sided t-test for two samples, Mean ± S.D.; n = 4 biological replicates). (E) Molecular combing of chromosomal DNA after a pulse of CldU for 30 min chased with a pulse of IdU for 30 min. Top: Schematic shows distances that were measured to estimate fork progression rate and inter-origin distance. Middle: Representative image of the combed DNA stained for CldU (green) and IdU (red) shown below the schematic. Bottom: Box and whiskers plot for fork progression rate and inter-origin distance of indicated clones of cells. (P value < 0.01, two-sided Wilcoxon rank sum test for two samples; N = number of tracks counted. P: Statistical significance of any difference between WT and ORC2-/- cells. (F) Overlap of the BrIP-seq peaks between WT and ORC2-/- cells. (G) Box and whiskers plot for inter-origin distances (measured by BrIP-seq) for each chromosome in WT and ORC2-/- cells. The median inter-origin distance for all chromosomes together indicated at bottom right. (H) Circos plot of Origins mapped by BrIP-seq for chromosome 1. Outer circle: the chromosome with the karytotyping bands. Inner two circles: the locations of BrIPseq peaks in the WT and ORC2-/- cell lines. (I) Distribution of BrIP-seq mapped origins relative to distance from Transcription Start Sites (TSS). In WT and ORC2-/- cells.

## CDC6 becomes more important for replication in *ORC1-/-* or *ORC2-/-* cells

The increase in chromatin association of CDC6 in the *ORC2-/-* cells (*Figure 1H*) led us to test whether CDC6 acts inefficiently to load enough MCM2-7 in the absence of the six-subunit ORC and the ATPase subunit of ORC (in the *ORC1-/-* cells). Knocking down CDC6 in WT cells (*Figure 5A*) did not decrease either the % of cells in active S phase (*Figure 5B*) or colony formation (*Figure 5C*). In contrast, knockdown of CDC6 in the *ORC1-/-* or *ORC2-/-* cells increased phosphoChk2, suggesting the activation of DNA damage checkpoints indicating problems in S phase (*Figure 5A*), decreased actively replicating cells, and decreased colony formation (*Figure 5B–C*), suggesting that CDC6 becomes more important for DNA replication and cell proliferation in the absence of ORC2 or ORC1.

## ORC5 protein remains important for DNA replication in *ORC1-/-* or *ORC2-/-* cells

To test whether any of the other subunits of the six-subunit ORC remain important for replication in the *ORC1-/-* or *ORC2-/-* cells, we depleted ORC5 by siRNA (*Figure 5D*). Knockdown of ORC5 did not change the cell cycle profile in *ORC-/-* mutant cells (*Figure 5E*), and in fact more severely repressed the % of actively replicating cells in WT cells than in the *ORC1-/-* or *ORC2-/-* cells (*Figure 5F*). However knockdown of ORC5 still inhibited BrdU incorporation in the *ORC1-/-* and *ORC2-/-* cells. We have not yet succeeded in knocking out the *ORC4* and *ORC5* genes, though of course such clones may emerge in future attempts. At the present state, our results suggest that the other subunits of ORC may support DNA replication independent of ORC1 or ORC2.

## Discussion

The surprising results from this genetic investigation of ORC in human cell lines suggest that two subunits of ORC are dispensable for DNA replication. Although we believe that no ORC2 protein is synthesized in the *ORC2-/-* cells, we have to entertain the caveat that up to 1530 ORC2 molecules can escape the limits of detection with our current antibodies. This number is still too few to license the 52,000 origins of replication mapped by two independent methods. It is unlikely that a single ORC complex can catalytically load thirty-five MCM2-7 hexamers distributed 100 kb apart over 3.5 Mb of chromosomal DNA.

Since knockdown of ORC5 still affects DNA synthesis in the *ORC1-/-* or *ORC2-/-* cells, we cannot yet conclude that replication initiation can occur in the absence of all subunits of ORC. To reach that conclusion we have to successfully delete both alleles of the other four subunits, *ORC3-6*. However, even if the remaining ORC subunits are functional for loading MCM2-7 our result requires reconsideration of existing models of ORC function in replication initiation. (1) The crystal structure suggests that ORC2-3-5-4-1 are arranged in a gapped ring (with a central channel that is wide enough to surround a DNA double-helix), and that later in licensing, CDC6 slips into the gap between ORC2 and ORC1 to close the gap. The ORC-CDC6 ring is proposed to interact with the MCM2-7 ring end-on-end during the loading of MCM2-7 (*Bleichert et al., 2015*). Loss of one subunit in the five-

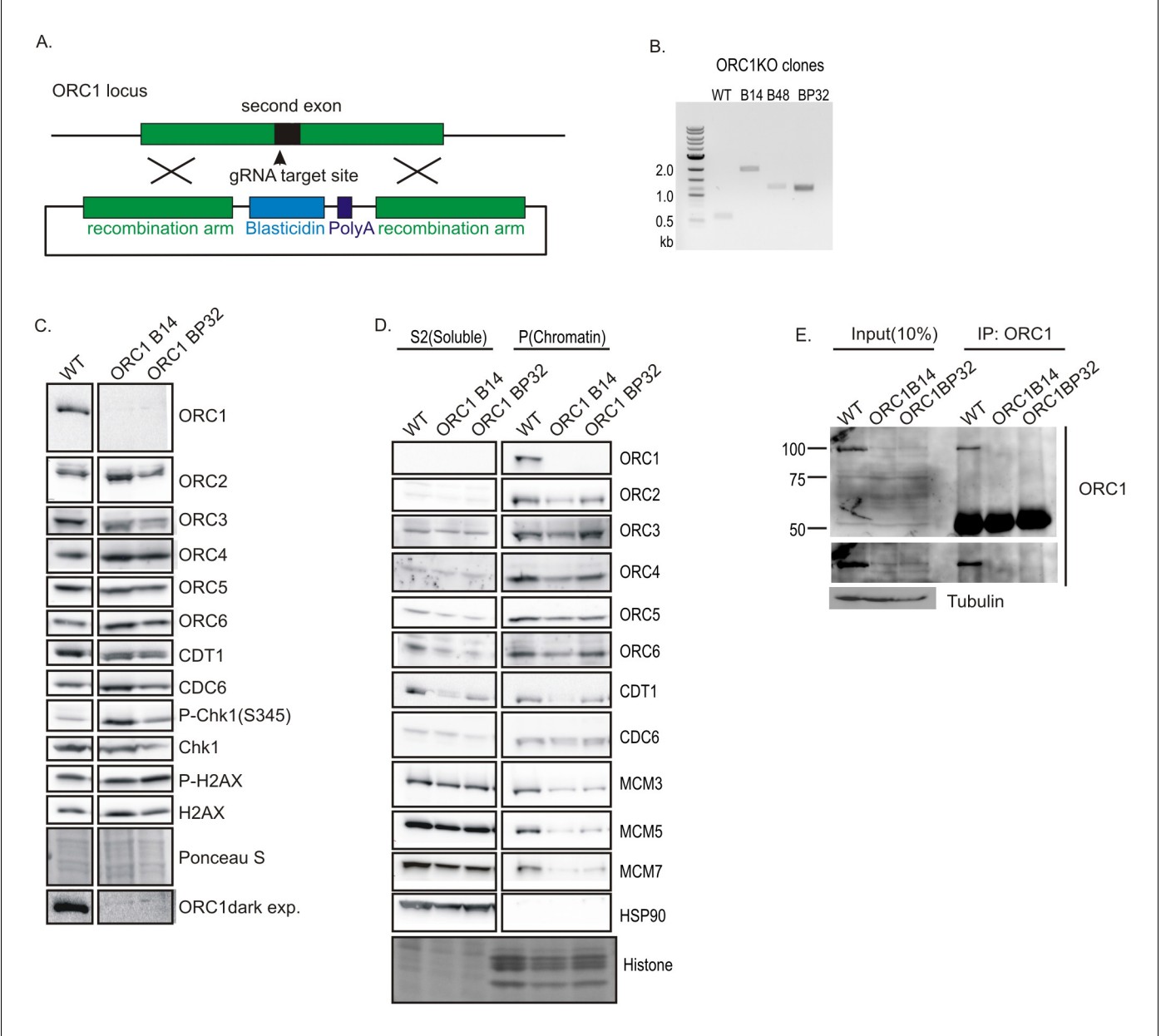

**Figure 3.** Knockout of ORC1 in HCT116 p53-/- cells. (**A**) Strategy for insertion of a blasticidin gene and poly A site after first methionine of ORC1 in the second exon. (**B**) PCR on genomic DNA of indicated clones. WT: HCT116 p53-/- and ORC1+/+. ORC1 Knockout clones, B14, B48 and BP32 have an insert on both alleles of ORC1 as indicated by the absence of 0.6 kb PCR product. (**C**) Western blot for indicated proteins in clones indicated on the top. Darker exposure of the ORC1 blots is shown at the bottom. Ponceau S stains all proteins on the blot and also indicates equal loading of lanes. (**D**) Immunoblot of soluble and chromatin-associated proteins in the clones indicated at the top. For each panel, all the lanes are from the same blot and exposure. (**E**) Input cell lysate and immunoprecipitates of ORC1 immunoblotted for ORC1. Darker exposure of the top blots is shown in the middle. Tubulin in the cell lysate or the IgG band in the immunoprecipitate serves as loading control.

membered ORC ring makes it difficult for the remaining subunits to form a ring large enough (i) to surround a DNA double-helix in the same manner as wild type ORC or (ii) to interact with the MCM2-7 ring end-on-end. (2) Human ORC1 and ORC4 are the only subunits that have intact Walker A and B motifs. Multiple groups have shown that the ATPase activity of ORC (in *S. cerevisiae*, in *D. melanogaster* and in *H. sapiens*) depends exclusively on the Walker A and B motifs of the ORC1 subunit, and that this ATP binding and hydrolysis activity is essential for ORC function

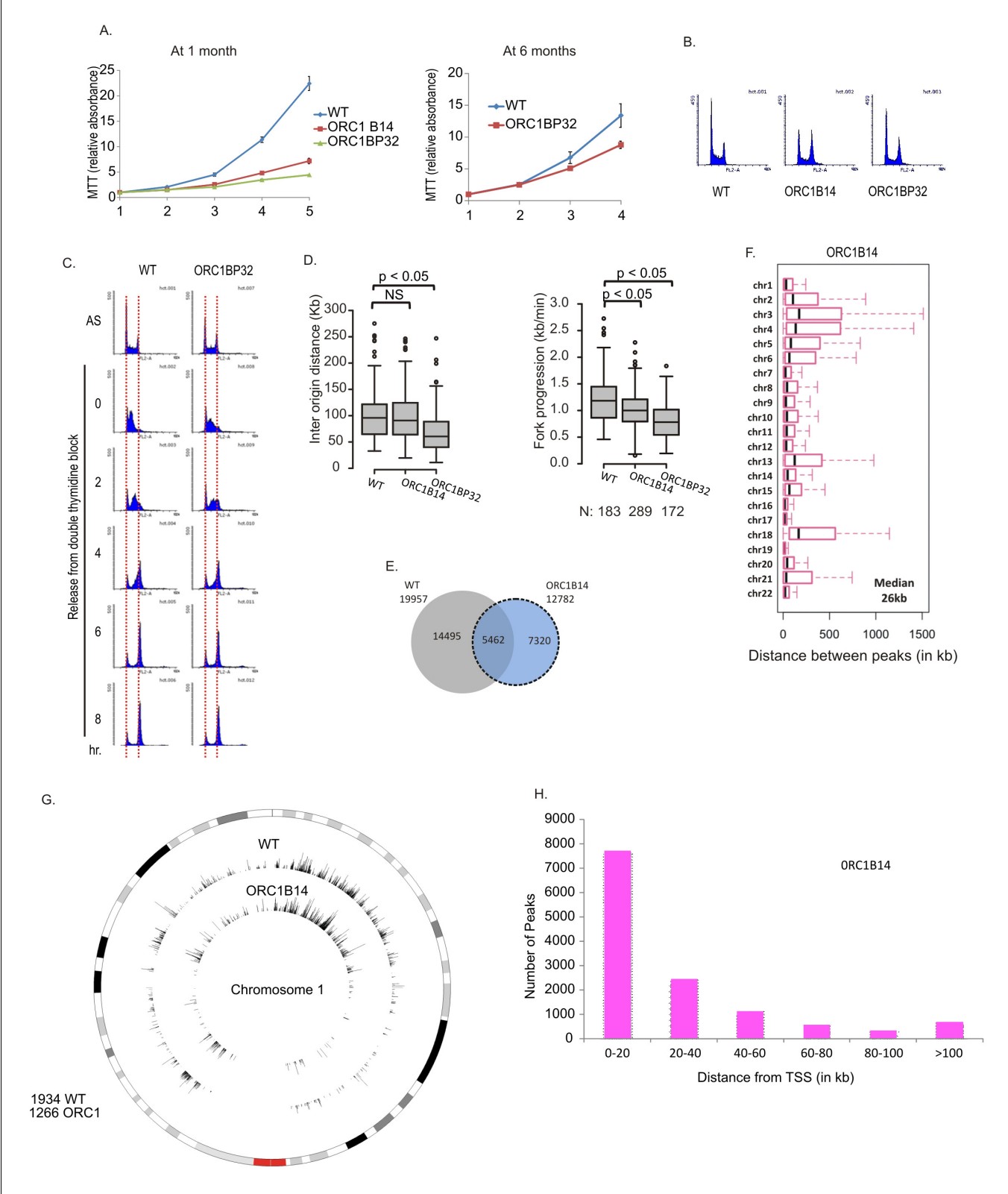

**Figure 4.** Cell proliferation and DNA replication changes in the ORC1-/- cell lines. (A) Growth curves of indicated clones of cells over four days, expressed as MTT absorbance relative to the level at day 1. (Mean ± S.D.; n = 4 biological replicates) Cells after passage for 1 month or six months. (B) FACS profile of propidium-iodide stained cell-cycle asynchronous cells from indicated clones. (C) Cells arrested in double-thymidine block released into nocodazole containing medium and harvested at indicated times after release to measure rate of progression through S phase. AS: asynchronous cells. *Figure 4 continued on next page*

*Figure 4 continued*

The red dotted lines indicate cells with G1 and G2 DNA content. (D) Molecular combing of chromosomal DNA after a pulse of CldU for 30 min chased with a pulse of IdU for 30 min. Box and whiskers plot for fork progression rate and inter-origin distance of indicated clones of cells. (P value < 4.6e-06, two-sided Wilcoxon rank sum test for two samples N = number of tracks counted) (Inter origin disntance N = 91(WT), 131(ORC1B14), 174(ORC1BP32) p: Statistical significance of any difference between WT and ORC1-/- cells. (E) Overlap of the BrIP-seq peaks between WT and ORC1-/- cells. (F) Box and whiskers plot for inter-origin distances (measured by BrIP-seq) for each chromosome in WT and ORC1-/- cells. The median inter-origin distance for all chromosomes together indicated at bottom right. (G) Circos plot of Origins mapped by BrIP-seq for chromosome 1. Outer circle: the chromosome with the karyotyping bands. Inner two circles: the locations of BrIPseq peaks in the WT and ORC1-/- cell lines. (H) Distribution of BrIP-seq mapped origins in ORC1-/- cells relative to distance from Transcription Start Sites (TSS).

(*Chesnokov et al., 2001*; *Giordano-Coltart et al., 2005*; *Klemm et al., 1997*). Even if an altered or partial ORC is initiating replication, we have to conclude that any ATPase activity necessary can be provided by ORC4 or CDC6.

There is a report arguing that ORC1 is not essential for endoreplication in Drosophila, because *ORC1-/-* larvae still allowed endoreplication in salivary gland cells, with only a twofold reduction of ploidy (*Park and Asano, 2008*). However the paper did not show the sensitivity of the western-blot to measure the level of residual ORC, so that it is theoretically possible that there was enough residual maternal ORC in the endoreplicating cells.

The classic model of replication initiation where ORC first associates with the DNA, helps load CDC6, which then helps load CDT1 and MCM2-7 may still be important for efficient MCM2-7 loading. The surprise is that inefficient MCM2-7 loading, perhaps with the help of CDC6, is sufficient for replication initiation and cell survival in the absence of six-subunit ORC. The other surprise is that the six-subunit ORC does not appear to associate with chromatin as a holocomplex. Clearly ORC6 associates with chromatin normally despite a decrease in ORC1 (in the *ORC2-/-* or *ORC1-/-* cells) or ORC2-5 loading (in the *ORC2-/-* cells). Similarly ORC2-5 loading is independent of ORC1 loading (in the *ORC1-/-* cells).

The origin plasticity of eukaryotes is attributed to the loading of excess subunits of MCM2-7 on chromatin. It is thus also surprising that the plasticity persists despite a significant reduction in the association of MCM2-7 on chromatin. In summary, we suggest that the absolute requirement for six-subunit ORC for licensing bi-directional origins of replication can be bypassed in some cell lines.

## Materials and methods

### Cell culture and transfection

HCT116 p53−/− cells (*Bunz et al., 1998*) (RRID:CVCL_S744, a generous gift from Fred Bunz, Johns Hopkins) were maintained in McCoy's 5A-modified medium supplemented with 10% fetal bovine serum. HBEC3 (RRID:CVCL_X491)-p53KD-K-RasV12 cells (a generous gift from, David R. Jones, Memorial Sloan-Kettering Cancer Center) were originally engineered to express sh-p53 RNA and K-Rasv12 protein by John D. Minna's group in the University of Texas Southwestern Medical Center (*Sato et al., 2006*) and were maintained in Keratinocyte-SFM. HBEC3-p53KD-K-RasV12 cells we used also express shGL2 and GFP (*Hall et al., 2014*). Lipofectamine 2000 (Thermo Fisher Scientific, Waltham, MA) was used to transfect plasmids and RNAiMAX (Thermo Fisher Scientific) was used to transfect siRNAs according to the manufacturer's protocol. CDC6 siRNA (GAUCGACUUAAUCAGGUAU), ORC5 siRNA (CCCUGGUUGGCCAUGACGA) was synthesized by Thermo Fisher Scientific, Waltham, MA. HEK293T cells (RRID:CVCL_0063) were from ATCC (CRL-3216). No mycoplasma contamination was found. 293T cells and HCT116 p53-/- cells were authenticated by STR profiling.

### Plasmids

gRNA was cloned into pCR-Blunt II-TOPO vector backbone (Addgene 41820, Cambridge, MA) using PCR and In-Fusion cloning (Clontech, Mountain View, CA). gRNA target sequence was as follows. GAAGGAGCGAGCGCAGCTTTTGG. Human codon optimized Cas9 nuclease (hCas9) expression vector was obtained from Addgene (41815).

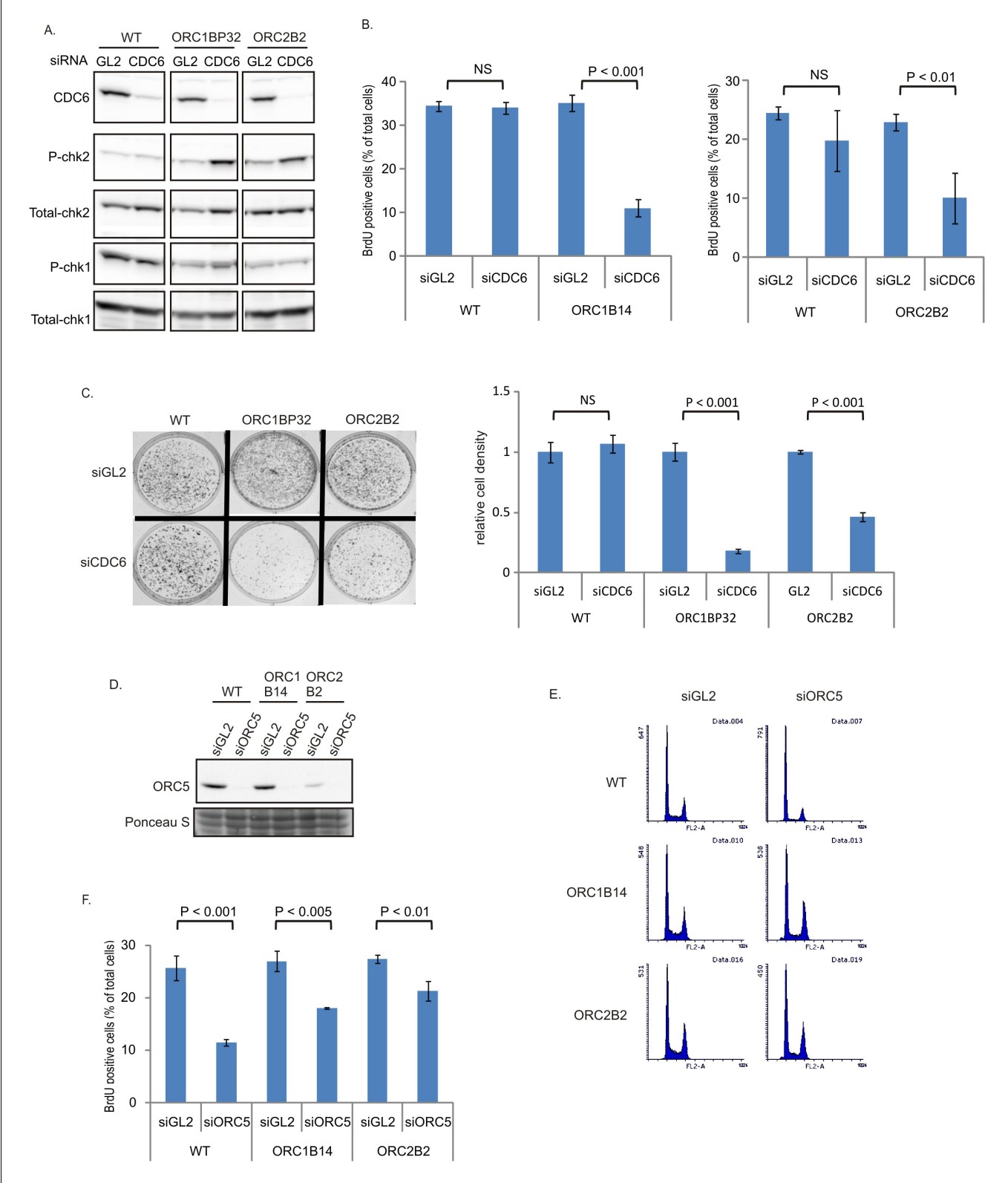

**Figure 5.** CDC6 is more essential for replication and colony formation in the ORC mutant cells. (**A**) Immunoblots of extracts from indicated cell lines following transfection of siGL2 (negative control siRNA against luciferase) or siCDC6. (**B**) % of BrdU+ cells after transfection of indicated siRNAs. Data from two color FACS. (P value < 0.01, two-sided t-test for two samples, Mean ± S.D. n = 4 or 3 biological replicates) (**C**) Top: 72 hr after transfection of indicated siRNAs, 2000 cells were plated per plate for colony formation detected by Crystal violet staining after seven days. Bottom: Crystal violet

*Figure 5 continued on next page*

*Figure 5 continued*
stained colony density were measured. Data presented for each cell line normalized to the density of the siGL2 transfected cells. (P value < 0.001, two-sided t-test for two samples, Mean ± S.D. n = 3 biological replicates). (**D**) Immuno blots of ORC5 (**E**) FACS profile of propidium-iodide stained cells for cell-cycle determination at three days after transfection of indicated siRNA. (**F**) % of BrdU+ cells after transfection of indicated siRNAs. Data from two color FACS. (Two-sided t-test for two samples, Mean ± S.D. n = 3 biological replicates).

## Construction of vectors for homologous recombination

Blasticidin or Hygromycin resistance genes terminated by a polyA sequence and flanked by two homology arms (0.9 kb–1.6 kb in length) were cloned into pDONR 221 (Thermo Fisher Scientific) using PCR and In-Fusion cloning (Clontech).

## DNA combing assay

Cells were pulse labeled with 100 µM CldU for 30 min, following by 250 µM IdU for 30 min before embedding into agarose plug. DNA was combed on silanized coverslips (Genomic Vision, Bagneux, France), dehydrated at 65°C for 4 hr and denatured in 0.5 M NaOH and NaCl for 8 min. CldU or IdU were immune-detected with either anti-BrdU antibody that recognizes CldU (MA 182088, Thermo Fisher Scientific, Waltham, MA, RRID:AB_927214) or anti-BrdU antibody that recognizes IdU (347580, BD Biosciences, Franklin Lakes, NJ, RRID:AB_400326). Image acquisition was performed with Zeiss AxioObserver Z1, 63 X objective. DNA lengths were measured using Image J software.

## BrdU incorporation

BrdU incorporation was conducted as previously described (*Machida et al., 2005*) with the following modifications. Cells were labeled with 10 µM BrdU for 30 min and fixed in 70% Ethanol. DNA was denatured in 2 M hydrochloric acid and stained with FITC-conjugated BrdU antibody (556028, BD Biosciences, RRID:AB_396304) and propidium iodide (Sigma-Aldrich, St.Louis) according to the manufacturer's instruction.

## Clonogenic assay

To determine the effects of CDC6 knock down, cells were transfected with siRNA twice. 48 hr after the first siRNA transfection, 2000 cells were plated in six well plates. Colonies were fixed, stained with crystal violet, and counted 1 week later. All experiments were conducted in triplicate.

## Proliferation assay

1000 cells were plated in 96 well plates. The absorbance of cells were measured every 24 hr using CellTiter 96 Non-Radioactive Cell Proliferation Assay (Promega, Fitchburg, WI) according to the manufacturer's instructions. All experiments were conducted in triplicate and absorbance relative to that on day one was expressed.

## Immunoprecipitation, western blot, antibodies, and recombinant protein

Cells were lysed in lysis buffer 50 mM Tris-HCl (pH 8.0), 150 mM NaCl, 5 mM EDTA, 0.5 % NP-40, 1 mM DTT, 20 mM NaF, and protease inhibitor cocktail. Lysate was cleared by centrifugation and incubated with ORC1(*Machida et al., 2005*) or ORC2 (*Dhar et al., 2001*) antibody for 4 hr. Immunoprecipitate was collected on Dynabeads Protein G (Thermo Fisher Scientific) and eluted with 2 x SDS sample buffer. Antibodies used were as follows. ORC2 (*Dhar et al., 2001*) (*Figures 1F, G, H,and and K*), ORC2N (*Figures 1C, D,and and J*) (sc-32734, Santa Cruz Biotechnology, Dallas, TX, RRID: AB_2157726), ORC2C (*Figure 1C and E*) (sc-13238, Santa Cruz Biotechnology, RRID:AB_2157715), MCM3 (sc-9850, Santa Cruz Biotechnology, RRID:AB_2142269), HSP90 (sc-13119, Santa Cruz Biotechnology, RRID:AB_675659), Cdt1 (*Senga et al., 2006*), ORC3, ORC4, ORC5, and ORC6 (*Machida et al., 2005*), ORC1 (4731, Cell Signaling Technology, Danvers, MA, RRID:AB_2157583), CDC6 (3387, Cell Signaling Technology, RRID:AB_2078525), p-Chk1 (2341, Cell Signaling Technology, RRID:AB_330023), p-Chk2 (2661, Cell Signaling Technology, RRID:AB_331479), Chk2 (3440, Cell Signaling Technology, RRID:AB_2229490), p-H2AX (2577, Cell Signaling Technology, RRID:AB_

2118011), and H2AX (2595, Cell Signaling Technology, RRID:AB_10694556), MCM5 (A300-195A, Bethyl Laboratories, Inc Montgomery, TX, RRID:AB_185552), MCM7 (A300-128A, Bethyl Laboratories, Inc., RRID:AB_2142821), FLAG (F1804, Sigma, RRID:AB_262044), and Chk1 (NB100-464, Novus Biologicals, LLC, Littleton, CO, RRID:AB_10002158). GST tagged ORC2 recombinant protein was purchased (H00004999-P01, Abnova, Taipei City, Taiwan)

### Br-IP Seq

The cells were labelled with 100 μM BrdU (Sigma) in exponential phase of their growth (50–60% of confluency) for 1 hr. The cells were lysed and genomic DNA was isolated, denatured and nascent strands were separated on neutral sucrose gradient. The fragments of 0.5 to 3.0 kb were selected. After dialysis against TE, the DNA was sheared, denatured and immune precipitated with anti-BrdU antibody (B8434, Sigma, RRID:AB_476811). The single stranded BrdU immunoprecipitate (10 ng) was then used to prepare next generation sequencing libraries using Takara Chip-Seq kit. The library from the control genomic DNA was prepared the same way as for the BrdU sample but the only difference was that sucrose gradient centrifugation and size selection was not done for the genomic control. Also the BrdU incorporation was for a longer time (36 hr compared to 1 hr for Br-IP sample). Single-end 75 bp reads were obtained for wildtype (WT) and ORC2 or ORC1 knockout cells. BrdU incorporated genomic strands was also sequenced and used as control (CNTL). Perl script was written and used to trim T's present at the 3' end of reads. The trimmed reads were aligned to hg38 using Bowtie2 with the default parameters. Alignment with Bowtie2 resulted in 12597288 (81%) and 10204177 (77%) and 13171609 (93%) mapped reads in WT, KO and CNTL respectively. To define peaks, the genome was divided into 1 kb windows. Any 1 kb window in WT or KO cells with two fold more reads than CNTL were considered to calculate mean and SD. 1 kb windows with reads $\geq$ mean + 3 SD number of reads were defined as peaks in WT or KO cells. Dataset for transcription start sites (TSSs) was downloaded from the UCSC genome browser. Circos v0.67 (*Krzywinski et al., 2009*) was used to construct circular genome visualizations. Peaks coordinates of WT and KO chromosome one was parsed to create files appropriately formatted for input to Circos.

### Chromatin fractionation

Chromatin fractionation was performed as previously described (*Méndez and Stillman, 2000*). Cells were resuspended in buffer A (10 mM HEPES [pH7.9], 10% glycerol, 1 mM DTT, protease inhibitor cocktail [Thermo Fisher]). After adding 0.1% Triton X-100, cells were incubated for 5 min on ice and centrifuge at 1300 g, 4°C. Supernatants were further clarified by centrifugation at 20000 g, 4°C (S2). Pellets (Nuclei) were washed in buffer A and lysed in buffer B (3 mM EDTA, 0.2 mM, EGTA, 1 mM DTT, protease inhibitor cockatil). After incubation for 30 min on ice, lysate was centrifuged at 1700 g, 4°C. Pellets (chromatin) were washed in buffer B and lysed in 2 x SDS sample buffer and sonicated (P3).

### Statistical analysis

We have used the pwr package in *R* for choosing sample size for all the experiments. Owing to the nature of the performed experiments, no randomization and no blinding were used. All test statistics were calculated with R (http://www.r-project.org/). t-test and Wilcoxon–rank sum test was performed to test the difference in mean for the normal and skewed data respectively.

### Data deposition

Source data including BrIP seq data were deposited in Dryad (*Shibata et al., 2016*).

## Acknowledgements

We thank Dutta lab members for the useful discussion. This study is supported by R01 CA060499 and CA166054 (to AD).

## Additional information

### Funding

| Funder | Grant reference number | Author |
|---|---|---|
| National Institutes of Health | CA060499 | Anindya Dutta |
| National Institutes of Health | CA166054 | Anindya Dutta |

The funders had no role in study design, data collection and interpretation, or the decision to submit the work for publication.

### Author contributions

ES, Conception and design, Acquisition of data, Analysis and interpretation of data, Drafting or revising the article; MK, Analysis and interpretation of data, Drafting or revising the article; YS, Acquisition of data, Analysis and interpretation of data, Drafting or revising the article; SS, SK, Acquisition of data, Drafting or revising the article; AD, Conception and design, Analysis and interpretation of data, Drafting or revising the article

### Author ORCIDs

Anindya Dutta, http://orcid.org/0000-0002-4319-0073

## Additional files

### Major datasets

The following dataset was generated:

| Author(s) | Year | Dataset title | Dataset URL | Database, license, and accessibility information |
|---|---|---|---|---|
| Shibata E, Kiran M, Shibata Y, Singh S, Kiran S, Dutta A | 2016 | Data from: Two subunits of human ORC are dispensable for DNA replication and proliferation. | http://dx.doi.org/10.5061/dryad.v84c2 | Available at Dryad Digital Repository under a CC0 Public Domain Dedication |

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
