## [Decision Letter]

Thank you for submitting your article "ORC is dispensable for DNA replication initiation, but essential for repression of Rb- and polycomb-regulated genes" for consideration by *eLife*. Your article has been reviewed by three peer reviewers, one of whom is a member of our Board of Reviewing Editors, and the evaluation has been overseen by Kevin Struhl as the Senior Editor. The reviewers have opted to remain anonymous.

The reviewers have discussed the reviews with one another and the Reviewing Editor has drafted this decision to help you prepare a revised submission.

Summary:

All reviewers find the work potentially important, but to validate that ORC is not essential for DNA replication additional work is required as stated below. Additionally, the experiments suggesting that the ORC complex is important for expression of Rb- and Polycomb target genes is based on single experiments. We suggest that you expand on these experiments or resubmit a shorter version focusing exclusively on the DNA replication part.

Essential revisions:

We all agree that the statement based on the deletion of ORC1 and ORC2, while suggestive of the conclusion, does not provide a definitive answer for ORC-independent DNA replication. We would like to see a deletion or knock down of ORC5.

We look forward receiving a modified version of the studies.

*Reviewer #1:*

In this manuscript, the authors aim to address the roles of ORC2 and ORC1 proteins in replication by knocking out these proteins in HCT116 and other cell lines. They observe no drastic effects on DNA replication, cell cycle progression, origin number and distribution. They also find that CDC6 plays an important role in this context and that a large number of PRC2 and Rb -dependent genes are deregulated.

Major comment:

Although these findings are important, they do not offer much novel insights. The authors have previously shown a number of the current findings in a hypomorphic ORC2 cell line (JBC (10), 6253-6260). Also, that ORC has roles in gene silencing also has been shown (Genes & Dev (9), 911-924 / Cell (91), 311-323). Additional data that provide some novel mechanistic insight into why ORC is dispensable for replication or what kind of role it plays in transcriptional regulation of PRC2 and Rb regulated genes are necessary to justify publication of this manuscript in *eLife*.

*Reviewer #2:*

The manuscript by Dutta and colleagues describes the analysis of deletions of ORC2 or ORC1, key subunits of the origin-recognition complex (ORC), in human cells. ORC is an essential replication protein in simple eukaryotes where origins of replication are defined by specific sequences. In contrast, higher eukaryotes lack a consensus sequence that defines origins and the molecular determinants of the sites of replication initiation in these organisms remain murky. The authors make the intriguing finding that deletions of either the ORC1 or the ORC2 gene are not lethal in human tissue culture cells. They go on to show that loading of the essential Mcm2-7 helicase occurs at reduced levels in these strains but residual loading is observed. Because the chromatin association of Cdc6 and Cdt1, two other helicase-loading proteins is unaffected, the authors suggest that continued association of these proteins mediates the residual Mcm2-7 loading. Intriguingly, depletion of Cdc6 reduced but did not eliminate replication in these cells (perhaps due to incomplete depletion).

The authors do a nice job of showing that deletions of ORC1 and ORC2 are viable and maintain relatively normal patterns of replication. The authors have also done well to show that their deletions are complete and there is not large amounts the proteins retained in the targeted cells. What is less clear is the method by which the residual Mcm2-7 is being loaded. Clearly Cdc6 is still playing a role but whether the remaining ORC subunits are capable of assisting with this is not clear. It would significantly strengthen the paper to show that siRNA of other ORC subunits does not have the same effect as there is either substantial (ORC1 deletion) or residual (ORC2 deletion) binding of other ORC subunits in each case. Overall, this part of the paper convincingly shows that there are other pathways than the canonical ORC-dependent ones to initiate replication.

The authors also observe that the loss of ORC causes changes in gene expression patterns that are similar to E2F1 over-expression strains, Rb knockdown strains and PRC2 subunit deletion strains. While these findings indicate that deletion of ORC subunits leads to changes in the patterns of gene expression, whether these changes are direct or indirect is unclear. The authors suggest that ORC is "critical for the regulation of these genes," however, they do not discuss the possibility that the changes in cell cycle progression as a possible reason for these changes. It would improve the paper to have a more complete discussion of the various reasons that these genes could be altered.

One other point that merits additional discussion is the selection that is going on as they grow up cells after deletion of ORC1 or ORC2. The authors point out that over time these cells are dividing faster which strongly suggests that there are additional genetic changes that alow improved cell growth. This means that the cells being analyzed are not simply *ORC1-/-* cells or *ORC2-/-* cells. It is important that the authors acknowledge this caveat and discuss what types of changes could be responsible for the improved growth.

Overall, the authors provide data that strongly suggests that there are non-canonical methods to initiate replication in the absence of ORC1 or ORC2. This is an important and surprising insight into the mechanism of initiation in mammalian cells and is consistent with previous data from this lab and in *Drosophila* that there could be ORC-independent mechanisms for replication initiation. It remains to be demonstrated whether the residual DNA replication observed is independent of the other ORC subunits (that is truly ORC- independent) or not (this would require double mutants in ORC – it would be a very nice addition if the authors showed that depletion/elimination of a second ORC subunit in the *ORC1-/-* or *ORC2-/-* cells did or did not further reduce replication). The gene expression data is relatively underdeveloped and the strong statements made by the authors about ORC being critical for their regulation seem premature without evidence that there is a direct mechanistic connection between ORC and the expression patterns observed rather than these changes being a general response to the altered cell cycle patterns observed when ORC is depleted.

*Reviewer #3:*

The authors present an interesting manuscript leading to four critical conclusions: 1. ORC proteins associate onto chromatin independently of their association as a holocomplex; 2. CDC6 and CDT1 are recruited to chromatin independently of ORC1 and ORC2; 3. enough MCM proteins are loaded to ensure a nearly undisrupted S-phase (albeit diminished cell proliferation) in the absence of ORC1 or ORC2; and 4. ORC proteins are important for the repression of Rb and PcG genes.

Using CRISPR-CAS9, neither ORC1 nor ORC2 deletions seemed to delay S-phase progression or affect the survival of the selected clones (passaged up to a year).

While ORC2-null cells modestly reduced the number of active origins of replication (~20% increase in distance between origins), ORC1-null cells did not show a loss of origin firing (perhaps even a slight increase). Unlike RNAi-based publications, ORC2 deletion did not lead to HP1 foci disruption or loss of chromatin condensation, but did delay mitosis and deregulate centrosomes.

The authors also present gene microarray and Gene Set Enrichment Analyses uncovering a global loss of silencing over genes regulated by Rb and PRC2 when either ORC1 or ORC2 are deleted.

While the claims are highly interesting, the data seems to scratch the surface when it comes to the relation between ORC, CDC6-CDT1 and MCM proteins. The conclusions that ORC proteins associate on chromatin without forming a complex, and that CDC6 and CDT1 deposit MCM proteins independently of ORC, nearly solely rely on a single western analysis of cells fractionated into soluble extracts and chromatin.

That the ORC proteins associate with chromatin independently of their association as a holocomplex can be further verified by IF, or by western analyses of the extracts separated by size (either through chromatography or on density gradients), for example. That CDC6 and CDT1 are recruited to chromatin independently of ORC is a strong claim and also needs further validation (IF analysis through cell division, etc.). Same goes for MCM deposition in the absence of a functional ORC complex (though that in itself could be a separate paper).

The manuscript should be strongly considered for publication, but only after the claims regarding ORC and CDC6-CDT1 association with chromatin in the absence of ORC are further investigated.

[Editors’ note: A previous version of this manuscript was rejected at revised stage, but the authors were invited to resubmit after an appeal against the decision.]

Thank you for submitting your work entitled "Human ORC is dispensable for DNA replication initiation, but essential for repression of retinoblastoma regulated genes." for consideration by *eLife*. Your article has been seen by the Reviewing Editor in consultation with one of the original reviewers and overseen by Kevin Struhl as the Senior Editor. The reviewer has opted to remain anonymous. Based on considerable discussion, we regret to inform you that your work will not be considered further for publication in *eLife*.

We recognize the importance and surprise to the field of the definitive demonstration that not all the ORC subunits are required for replication initiation or even cell viability. However, the concern expressed during the original review was whether ORC was dispensable (as implied in the original paper and would be a major shock) or whether individual subunits were dispensable. In response to the original reviews, the authors provided new data addressing the role of Orc5. Unfortunately, the results are not clear cut, and the authors simply state that they can't say that the other subunits are required or not. While not conclusive, the observed effects of Orc5 depletion as well as the failure to obtain knockouts of other Orc subunits suggest that a partial or altered ORC complex is necessary and sufficient for viability and replication initiation. If true, this result would be less interesting because many multiprotein complexes contain subunits that are not essential for the function of the complex. In any event, the limited information on how replication occurs in the absence of Orc1 or Orc2 renders this paper as an unexpected and interesting observation with little mechanistic understanding. In addition, the transcription data is very preliminary because there are many direct and indirect ways that gene expression changes can occur and this data is peripheral to the central point of the paper about the role of ORC in DNA replication. For these reasons, the work better suited for a more specialized journal.

[Editors' note: further revisions were requested prior to acceptance, as described below.]

Thank you for resubmitting your work entitled "Two subunits of human ORC are dispensable for DNA replication and proliferation" for further consideration at *eLife*. Your revised article has been favorably evaluated by Kevin Struhl (Senior and Reviewing editor) and one reviewer.

The manuscript has been improved but there is one remaining issue that needs to be addressed before acceptance. Specifically, the Orc5 data is interpreted in a way that masks the finding that Orc5 depletion is still important in the absence of Orc1 or Orc2 (last paragraph of the results). The current version concludes that Orc5 is no more important than what is observed when Orc1 and Orc2 is present and hat it does not become more important. The reviewer and I think that the correct conclusion is that Orc5 remains important even in the absence of the other factors, and this is further supported by the now unmentioned result that the authors have been unable to create viable deletions of other Orc subunits (recognizing that this could be a trivial result, but still in contrast to what is observe for Orc1 and Orc2). In fact, it would be useful for the paper to mention the failure to get viable deletions of other Orc subunits, of course with the above caveat. The latter interpretation supports the idea that at least one of the non-deleted subunits is still involved in replication (an important conclusion). If this issue is taken care of, the paper will be accepted.

---

## [Author Response]

*Summary:*

*All reviewers find the work potentially important, but to validate that ORC is not essential for DNA replication additional work is required as stated below. Additionally, the experiments suggesting that the ORC complex is important for expression of Rb- and Polycomb target genes is based on single experiments. We suggest that you expand on these experiments or resubmit a shorter version focusing exclusively on the DNA replication part.*

The influence of ORC on RB can be explained by Stillman’s recent paper in *eLife* (Hossain and Stillman, 2016. Opposing roles for DNA replication initiator proteins ORC1 and CDC6 in control of Cyclin E gene transcription. eLife, 5, pp. 10.7554/eLife.12785), which shows physical interaction between ORC1 and RB. So we have retained that result and cite the Stillman paper as a possible mechanism. We have removed the ORC control of PRC2 regulation, as suggested by you.

Essential revisions:

*We all agree that the statement based on the deletion of ORC1 and ORC2, while suggestive of the conclusion, does not provide a definitive answer for ORC-independent DNA replication. We would like to see a deletion or knock down of ORC5.*

We have been unable to delete both alleles of ORC4, 5, or 6. Since this is a negative result, we cannot say whether it says that these genes are essential, or whether we will eventually get a deletion after more attempts.

siRNA of ORC5 decreased BrdU incorporation (DNA synthesis) by 60% in WT cells, and only 30% in the *ORC1-/-* or *ORC2-/-* cells. Again we cannot conclude from this that ORC5 is non-essential because siRNA knockdown can still leave residual ORC5. This result has now been included in Figure 10 and discussed in the paper.

Please note that our paper is very careful to say that replication is independent of complete ORC (the origin recognition complex composed of 6 subunits). We are not (yet) claiming that all ORC subunits are non-essential for replication. This is now better stated in the paper.

*Reviewer #1:*

*In this manuscript, the authors aim to address the roles of ORC2 and ORC1 proteins in replication by knocking out these proteins in HCT116 and other cell lines. They observe no drastic effects on DNA replication, cell cycle progression, origin number and distribution. They also find that CDC6 plays an important role in this context and that a large number of PRC2 and Rb -dependent genes are deregulated.*

*Major comment:*

*Although these findings are important, they do not offer much novel insights. The authors have previously shown a number of the current findings in a hypomorphic ORC2 cell line (JBC (10), 6253-6260). Also, that ORC has roles in gene silencing also has been shown (Genes & Dev (9), 911-924 / Cell (91), 311-323). Additional data that provide some novel mechanistic insight into why ORC is dispensable for replication or what kind of role it plays in transcriptional regulation of PRC2 and Rb regulated genes are necessary to justify publication of this manuscript in eLife.*

We respectfully point out that our paper is the first to show that ORC has a role in gene silencing independent of its role in DNA replication. Jasper Rine’s 1995 paper was with ORC in *S. cerevisiae*. Michael Botchan’s 1997 paper was with ORC in *D. melanogaster*. In both the organisms ORC is also essential for replication initiation and so they were unable to conclude that ORC’s function in replication and gene regulation are independent activities. We show that the replication function and gene expression regulation functions of ORC are separable in mammalian cells. In addition, the positive results on transcription (that loss of ORC deregulates gene expression) provide a great positive control that attests to the successful depletion of functional ORC. This positive result is worth contrasting with the negative effects on DNA replication (that loss of ORC does not have much of an effect on DNA replication initiation).

We now refer to Stillman’s recent paper to cite a possible mechanism by which ORC could regulate RB (Hossain and Stillman, 2016. Opposing roles for DNA replication initiator proteins ORC1 and CDC6 in control of Cyclin E gene transcription. eLife, 5, pp. 10.7554/eLife.12785). ORC 1 protein reduction in the ORC1 KO or the ORC2 KO cells may explain the deregulation of RB regulated genes. We removed the transcriptional control of PRC2 regulation because we do not know the mechanism by which ORC regulates PRC2.

*Reviewer #2:*

*The manuscript by Dutta and colleagues describes the analysis of deletions of ORC2 or ORC1, key subunits of the origin-recognition complex (ORC), in human cells. ORC is an essential replication protein in simple eukaryotes where origins of replication are defined by specific sequences. In contrast, higher eukaryotes lack a consensus sequence that defines origins and the molecular determinants of the sites of replication initiation in these organisms remain murky. The authors make the intriguing finding that deletions of either the ORC1 or the ORC2 gene are not lethal in human tissue culture cells. They go on to show that loading of the essential Mcm2-7 helicase occurs at reduced levels in these strains but residual loading is observed. Because the chromatin association of Cdc6 and Cdt1, two other helicase-loading proteins is unaffected, the authors suggest that continued association of these proteins mediates the residual Mcm2-7 loading. Intriguingly, depletion of Cdc6 reduced but did not eliminate replication in these cells (perhaps due to incomplete depletion).*

*The authors do a nice job of showing that deletions of ORC1 and ORC2 are viable and maintain relatively normal patterns of replication. The authors have also done well to show that their deletions are complete and there is not large amounts the proteins retained in the targeted cells. What is less clear is the method by which the residual Mcm2-7 is being loaded. Clearly Cdc6 is still playing a role but whether the remaining ORC subunits are capable of assisting with this is not clear. It would significantly strengthen the paper to show that siRNA of other ORC subunits does not have the same effect as there is either substantial (ORC1 deletion) or residual (ORC2 deletion) binding of other ORC subunits in each case. Overall, this part of the paper convincingly shows that there are other pathways than the canonical ORC-dependent ones to initiate replication.*

The siORC5 result is now included in Figure 10. siRNA of ORC5 decreased BrdU incorporation (DNA synthesis) by 60% in WT cells, and only 30% in the *ORC1-/-* or *ORC2-/-* cells. Again we cannot conclude from this that ORC5 is non-essential because siRNA knockdown can still leave residual ORC5. This result has now been included in Figure 10 and discussed in the paper.

Our results show that replication is independent of complete ORC (The origin recognition complex composed of 6 subunits). We are not claiming that all ORC subunits are non-essential.

*The authors also observe that the loss of ORC causes changes in gene expression patterns that are similar to E2F1 over-expression strains, Rb knockdown strains and PRC2 subunit deletion strains. While these findings indicate that deletion of ORC subunits leads to changes in the patterns of gene expression, whether these changes are direct or indirect is unclear. The authors suggest that ORC is "critical for the regulation of these genes," however, they do not discuss the possibility that the changes in cell cycle progression as a possible reason for these changes. It would improve the paper to have a more complete discussion of the various reasons that these genes could be altered.*

There is not much difference in cell cycle progression in the ORC KO cells based on FACS profile or S-phase progression. (Figure 2, 7B, and C), therefore we do not think that a change in cell cycle progression is responsible for the gene expression change. In addition, we now cite Stillman’s recent paper in *eLife* for a possible explanation of how ORC could regulate RB targeted genes.

*One other point that merits additional discussion is the selection that is going on as they grow up cells after deletion of ORC1 or ORC2. The authors point out that over time these cells are dividing faster which strongly suggests that there are additional genetic changes that alow improved cell growth. This means that the cells being analyzed are not simply ORC1-/- cells or ORC2-/- cells. It is important that the authors acknowledge this caveat and discuss what types of changes could be responsible for the improved growth.*

We have acknowledged this caveat. We do not know what change (genetic, epigenetic, or post transcriptional) occurred in *ORC1-/-* cells as they were passaged. In the future, we plan to do to whole genome sequencing, RNA sequencing, and Mass spec analysis from ORC1 KO cells in early and late passage to determine the mechanism of this adaptation. This is now discussed.

*Overall, the authors provide data that strongly suggests that there are non-canonical methods to initiate replication in the absence of ORC1 or ORC2. This is an important and surprising insight into the mechanism of initiation in mammalian cells and is consistent with previous data from this lab and in Drosophila that there could be ORC-independent mechanisms for replication initiation. It remains to be demonstrated whether the residual DNA replication observed is independent of the other ORC subunits (that is truly ORC- independent) or not (this would require double mutants in ORC – it would be a very nice addition if the authors showed that depletion/elimination of a second ORC subunit in the ORC1-/- or ORC2-/- cells did or did not further reduce replication). The gene expression data is relatively underdeveloped and the strong statements made by the authors about ORC being critical for their regulation seem premature without evidence that there is a direct mechanistic connection between ORC and the expression patterns observed rather than these changes being a general response to the altered cell cycle patterns observed when ORC is depleted.*

Discussed above.

*Reviewer #3:*

*[…]That the ORC proteins associate with chromatin independently of their association as a holocomplex can be further verified by IF, or by western analyses of the extracts separated by size (either through chromatography or on density gradients), for example. That CDC6 and CDT1 are recruited to chromatin independently of ORC is a strong claim and also needs further validation (IF analysis through cell division, etc.). Same goes for MCM deposition in the absence of a functional ORC complex (though that in itself could be a separate paper).*

We have performed immunolfuorescence of MCM5 after extraction of nucleiplasmic proteins and confirmed the lower intensity of MCM5 staining in ORC1 or ORC2 KO cells. These data are now included as Figure 1—figure supplement 5.

We can co-IP ORC2 and ORC3 in the *ORC1 -/-* cells (data not shown), but neither this experiment nor the experiment the reviewer suggests addresses whether the chromatin association of MCM2-7 is effected by an ORC sub complex, or by individual ORC subunits. To test whether interactions between the residual ORC subunits (ORC subcomplex formation) is necessary for the residual ORC or MCM loading, we have to knock in mutations that disrupt interactions between the remaining ORC subunits and test the phenotype. As the reviewer suggests, that would clearly be a separate paper.

[Editors' note: the author responses to the re-review follow.]

We recognize the importance and surprise to the field of the definitive demonstration that not all the ORC subunits are required for replication initiation or even cell viability. However, the concern expressed during the original review was whether ORC was dispensable (as implied in the original paper and would be a major shock) or whether individual subunits were dispensable. In response to the original reviews, the authors provided new data addressing the role of Orc5. Unfortunately, the results are not clear cut, and the authors simply state that they can't say that the other subunits are required or not. While not conclusive, the observed effects of Orc5 depletion as well as the failure to obtain knockouts of other Orc subunits suggest that a partial or altered ORC complex is necessary and sufficient for viability and replication initiation. If true, this result would be less interesting because many multiprotein complexes contain subunits that are not essential for the function of the complex. In any event, the limited information on how replication occurs in the absence of Orc1 or Orc2 renders this paper as an unexpected and interesting observation with little mechanistic understanding. In addition, the transcription data is very preliminary because there are many direct and indirect ways that gene expression changes can occur and this data is peripheral to the central point of the paper about the role of ORC in DNA replication. For these reasons, the work better suited for a more specialized journal.

Thank you for the decision letter and for talking with me about the issues raised by the reviewers. I appreciate the fact that the reviewers believe that the cells are viable and replicating in the absence of Orc2 or Orc1, but are concerned that we have not ruled out that a partial or altered subcomplex of the residual subunits is somehow eking out replication. This is my appeal asking for a re-consideration. If you agree, we can incorporate these points in the discussion.

1) I agree that we cannot rule out the possibility that a partial or altered subcomplex of the residual ORC subunits (comprising perhaps Orc3-4-5-6) carries out the function of the six- subunit ORC. However, I believe that such a possibility still makes us reconsider how ORC functions. Jim Berger’s beautiful structure of ORC (1) makes the following suggestions:

a) Orc2-3-5-4-1 are arranged in a gapped ring (in that order) with a central channel of 20A that is wide enough to surround a DNA double-helix, and that later in licensing, Cdc6 slips into the gap between Orc2 and Orc1 to close the gap (The schematic from their paper shows the ring of the Winged-Helix domains of ORC subunits). The ORC-Cdc6 ring is proposed to interact with the MCM2-7 ring end-on-end during the loading of MCM2-7. Loss of Orc2 or of Orc1 makes it difficult for the remaining subunits to form a ring large enough (i) to surround a DNA double-helix in the same manner as wild type ORC or (ii) to interact with the MCM2-7 ring end-on-end.

b)The structure of ORC has two tiers of rings offset with respect to each other, a tier of Winged-Helix (WH) domains and a tier of AAA+ domains. The WH domain of Orc1 interacts with the AAA+ domain of Orc4, and the AAA+ like domain of Orc2 interacts with the WH domain of Orc3. One could suggest that another cellular molecule substitutes for the missing Orc subunit in the two-tiered ring. However, the alternate molecule that replaces Orc1 (in the *Orc1-/-* cells) or Orc2 (in the *Orc2-/-* cells) not only has to have a WH domain and an AAA+ domain in a similar configuration as Orc subunits, but also has to retain the specific interactions that allow the Orc1WH-Orc4AAA or the Orc2AAA-Orc3WH interactions mentioned above.

c) Cdc6 is proposed to slip into the gap between Orc2 and Orc1 to close the ring and lock the DNA in place. Deletion of Orc2 or Orc1 creates problems for that model, because Cdc6 interacts with the Orc2 and Orc1 subunits of the open ring to close the latter.

d) The AAA+ domain of Orc1 blocked the central channel in the crystal structure and it was proposed that this is an auto-inhibited form of ORC that will be activated at the correct time in the cell-cycle to allow the DNA to slip into place in the central channel. Such autoinhibition is clearly not necessary because replication can proceed independent of Orc1.

2) Biochemical analyses of human ORC formation carried out 15 years back in my lab and that of others (2-4) proposed that Orc2 together with Orc3 is the core of the complex (the Orc2-3 dimer forms easily without other subunits). The Orc2-3 complex can associate with Orc5 but not Orc4. The Orc2-3-5 complex brings in Orc4, and finally the Orc2-3-5-4 complex associates with Orc1. This order of subunits entering the complex (Orc2-3-5-4-1) exactly recapitulates the order of the subunits around the ring in the Berger structure. It is hard to imagine how a residual complex can be formed in the absence of Orc2.

3) Human Orc1 and Orc4 are the only subunits that have intact Walker A and B motifs. The other subunits (Orc2, 3 and 5) have AAA+-like domains but not intact Walker A and B motifs. Multiple groups have shown that the ATPase activity of ORC (in *S. cerevisiae*, in *D. melanogaster* and in *H. sapiens*) depends *exclusively* on the Walker A and B motifs of the Orc1 subunit, and that this ATP binding and hydrolysis activity is essential for ORC function (5-7). For example, Orc1 (and the Walker A motif of Orc1) is essential for human ORC to bind to chromatin and to support DNA replication in *Xenopus* egg extracts (6). Thus has risen the model that the ATP binding and hydrolysis by the Orc1 subunit of ORC is critical for replication licensing. This model has to be reconsidered now that we show that cells are viable without Orc1, even if an altered or partial ORC is initiating replication. If an alternate cellular molecule substitutes for Orc1 in this critical function, it is most likely Cdc6. This is why we like the result that the Cdc6 gene becomes essential for cell proliferation in the cells deleted for Orc1 (or Orc2).

4) The BAH (Bromo-Adjacent-Homology) domain of Orc1 has been proposed to interact with nucleosomes, and this interaction deemed to be essential for origin selection and replication (8,9). A mutation in the BAH domain of Orc1 leads to Meier-Gorlin syndrome. Even if Cdc6 were to substitute for Orc1, Cdc6 does not have a BAH domain. Thus the importance of ORC- nucleosome interaction in origin specification has to be reconsidered.

5) When we have such a surprising result on the replication front, it is comforting to know that the at least another proposed function of ORC (regulation of gene expression) is evident in the *Orc1-/-* or *Orc2-/-* cells. The robust and reproducible changes in the gene expression program assure the reader that we have affected something in the cell upon deletion of Orc1 or Orc2. That is why we would like to keep it in the paper, but will delete it if you make that a pre- condition for accepting the paper in *eLife*. We think the regulation of Rb/E2F genes by ORC is explained by what Stillman has just published in 2016 in *eLife* (10).

6) There is a paper in EMBO J. suggesting that Orc2 is essential for properly condensed mitotic chromosomes and chromosome congression in the mitotic plate (11). This is why we thought it useful to show that Orc2 deletion had no effect on these phenotypes. Also, if there is residual DNA damage in cells replicating in the absence of Orc2, we would expect a severe block to entry into M phase (because of activation of the S to M checkpoint). This is why we examined entry and passage through mitosis of these cells. We can take out all this data if you make it a precondition for acceptance in *eLife*.

7) There is a paper in Science suggesting that Orc1 knockdown leads to 25-40% of cells having multiple centrosomes, compared to 2.5% of control cells (12). (From the abstract: “We report a new role for the Orc1 protein, a subunit of the Origin Recognition Complex (ORC) that is a key component of the DNA replication licensing machinery in controlling centriole and centrosome copy number in human cells, independent of its role in DNA replication.”) This is why we show that *Orc2-/-* cells barely increase centrosome number compared to WT cells (from 10% to 15%). We have similar data for *Orc1-/-* cells but did not include it in the paper. We can leave (or add to) this data in the paper, but can take it out if you make it a precondition for acceptance in *eLife*.

References:

1. Bleichert F, Botchan MR, Berger JM. Crystal structure of the eukaryotic origin recognition complex. Nature. 2015;519(7543):321-6. doi: 10.1038/nature14239. PubMed PMID: 25762138; PMCID: PMC4368505.

2. Dhar SK, Delmolino L, Dutta A. Architecture of the human origin recognition complex. J Biol Chem. 2001;276(31):29067-71. PubMed PMID: 11395502.

3. Vashee S, Simancek P, Challberg MD, Kelly TJ. Assembly of the human origin recognition complex. J Biol Chem. 2001;276(28):26666-73. doi: 10.1074/jbc.M102493200. PubMed PMID: 11323433.

4. Siddiqui K, Stillman B. ATP-dependent assembly of the human origin recognition complex. J Biol Chem. 2007;282(44):32370-83. doi: 10.1074/jbc.M705905200. PubMed PMID: 17716973.

5. Chesnokov I, Remus D, Botchan M. Functional analysis of mutant and wild-type *Drosophila* origin recognition complex. Proc Natl Acad Sci U S A. 2001;98(21):11997-2002. doi: 10.1073/pnas.211342798. PubMed PMID: 11593009; PMCID: PMC59756.

6. Giordano-Coltart J, Ying CY, Gautier J, Hurwitz J. Studies of the properties of human origin recognition complex and its Walker A motif mutants. Proc Natl Acad Sci U S A. 2005;102(1):69-74. doi: 10.1073/pnas.0408690102. PubMed PMID: 15618391; PMCID: PMC544074.

7. Klemm RD, Austin RJ, Bell SP. Coordinate binding of ATP and origin DNA regulates the ATPase activity of the origin recognition complex. Cell. 1997;88(4):493-502. PubMed PMID: 9038340.

8. Kuo AJ, Song J, Cheung P, Ishibe-Murakami S, Yamazoe S, Chen JK, Patel DJ, Gozani O. The BAH domain of ORC1 links H4K20me2 to DNA replication licensing and Meier-Gorlin syndrome. Nature. 2012;484(7392):115-9. doi: 10.1038/nature10956. PubMed PMID: 22398447; PMCID: PMC3321094.

9. Muller P, Park S, Shor E, Huebert DJ, Warren CL, Ansari AZ, Weinreich M, Eaton ML, MacAlpine DM, Fox CA. The conserved bromo-adjacent homology domain of yeast Orc1 functions in the selection of DNA replication origins within chromatin. Genes Dev. 2010;24(13):1418-33. doi: 10.1101/gad.1906410. PubMed PMID: 20595233; PMCID: PMC2895200.

10. Hossain M, Stillman B. Opposing roles for DNA replication initiator proteins ORC1 and CDC6 in control of Cyclin E gene transcription. *ELife*. 2016;5. doi: 10.7554/*eLife*.12785. PubMed PMID: 27458800; PMCID: PMC4987141.

11. Prasanth SG, Prasanth KV, Siddiqui K, Spector DL, Stillman B. Human Orc2 localizes to centrosomes, centromeres and heterochromatin during chromosome inheritance. EMBO J. 2004;23(13):2651-63. doi: 10.1038/sj.emboj.7600255. PubMed PMID: 15215892; PMCID: PMC449767.

12. Hemerly AS, Prasanth SG, Siddiqui K, Stillman B. Orc1 controls centriole and centrosome copy number in human cells. Science. 2009;323(5915):789-93. doi: 10.1126/science.1166745. PubMed PMID: 19197067; PMCID: PMC2653626.

[Editors' note: further revisions were requested prior to acceptance, as described below.]

Thank you for responding favorably to our appeal and allowing us to submit the paper as a Short Report. You make it clear that:

“It should focus on the key result (i.e. no transcriptional experiments) and a discussion of how the results are interpreted in terms of current knowledge. In particular, the discussion should explain how the results challenge the current view of ORC in terms of structural and biochemical knowledge, not a claim (not accepted by reviewers) that replication can occur without ORC.”

Below is a point-by-point dispensation of these issues.:

1) “No transcriptional experiments”: All results other than those pertaining to DNA replication have been removed.

2) “Not a claim (not accepted by reviewers) that replication can occur without ORC”: Done. We have even changed the title to say that “Two subunits of human ORC are dispensable…”.

3) “Focus on how the results challenge the current view of ORC”: We have done this. The Discussion is re-written to say that missing one subunit of a six-subunit ring that (a) encircles DNA and (b) interacts with MCM2-7 end-to-end, produces a spatial problem that must be somehow partially overcome in these cells. We also point out that the results show that all six subunits of ORC do not need to associate with chromatin in human cells as a holocomplex and that survival in the absence of ORC1 either suggests that the ATPase activity of ORC can be provided by ORC4 or CDC6, unlike the existing notion that ORC1 is the only subunit responsible for the ATPase activity of ORC.